# Metabolomic Profiling in Patients with Different Hemodynamic Subtypes of Severe Aortic Valve Stenosis

**DOI:** 10.3390/biom13010095

**Published:** 2023-01-03

**Authors:** Philipp Bengel, Manar Elkenani, Bo E. Beuthner, Maik Pietzner, Belal A. Mohamed, Beatrix Pollok-Kopp, Ralph Krätzner, Karl Toischer, Miriam Puls, Andreas Fischer, Lutz Binder, Gerd Hasenfuß, Moritz Schnelle

**Affiliations:** 1Clinic for Cardiology & Pneumology, University Medical Center Göttingen, 37075 Göttingen, Germany; 2DZHK (German Centre for Cardiovascular Research), Partner Site Göttingen, 37075 Göttingen, Germany; 3MRC Epidemiology Unit, University of Cambridge, Cambridge CB2 0QQ, UK; 4Computational Medicine, Berlin Institute of Health at Charité–Universitätsmedizin Berlin, 10117 Berlin, Germany; 5Department of Transfusion Medicine, University Medical Center Göttingen, 37075 Göttingen, Germany; 6Department of Pediatrics and Adolescent Medicine, Division of Pediatric Neurology, University Medical Center Göttingen, 37075 Göttingen, Germany; 7Department of Clinical Chemistry, University Medical Center Göttingen, 37075 Göttingen, Germany; 8Division Vascular Signaling and Cancer, German Cancer Research Center, 69120 Heidelberg, Germany

**Keywords:** severe aortic valve stenosis, hemodynamic subgroups, heart failure, metabolomics, metabolic remodeling

## Abstract

Severe aortic stenosis (AS) is a common pathological condition in an ageing population imposing significant morbidity and mortality. Based on distinct hemodynamic features, i.e., ejection fraction (EF), transvalvular gradient and stroke volume, four different AS subtypes can be distinguished: (i) normal EF and high gradient, (ii) reduced EF and high gradient, (iii) reduced EF and low gradient, and (iv) normal EF and low gradient. These subtypes differ with respect to pathophysiological mechanisms, cardiac remodeling, and prognosis. However, little is known about metabolic changes in these different hemodynamic conditions of AS. Thus, we carried out metabolomic analyses in serum samples of 40 AS patients (n = 10 per subtype) and 10 healthy blood donors (controls) using ultrahigh-performance liquid chromatography–tandem mass spectroscopy. A total of 1293 biochemicals could be identified. Principal component analysis revealed different metabolic profiles in all of the subgroups of AS (All-AS) vs. controls. Out of the determined biochemicals, 48% (n = 620) were altered in All-AS vs. controls (*p* < 0.05). In this regard, levels of various acylcarnitines (e.g., myristoylcarnitine, fold-change 1.85, *p* < 0.05), ketone bodies (e.g., 3-hydroxybutyrate, fold-change 11.14, *p* < 0.05) as well as sugar metabolites (e.g., glucose, fold-change 1.22, *p* < 0.05) were predominantly increased, whereas amino acids (e.g., leucine, fold-change 0.8, *p* < 0.05) were mainly reduced in All-AS. Interestingly, these changes appeared to be consistent amongst all AS subtypes. Distinct differences between AS subtypes were found for metabolites belonging to hemoglobin metabolism, diacylglycerols, and dihydrosphingomyelins. These findings indicate that relevant changes in substrate utilization appear to be consistent for different hemodynamic subtypes of AS and may therefore reflect common mechanisms during AS-induced heart failure. Additionally, distinct metabolites could be identified to significantly differ between certain AS subtypes. Future studies need to define their pathophysiological implications.

## 1. Introduction

Severe aortic valve stenosis (AS) represents the most prevalent valvular heart disease constituting a major driver of mortality and morbidity in the elderly population within the western world [1]. Over the last years transcatheter aortic valve replacement (TAVR) has been established as the therapy of choice for AS patients at high, moderate, as well as low surgical risk [1,2,3,4]. Although TAVR constitutes the causal therapy of the underlying disease, long-term prognosis in distinct patient populations remains poor [5,6]. Amongst other factors, prognosis following TAVR largely depends on the respective hemodynamic subtype. Current guidelines determine four different subtypes of AS that differ in aortic valve gradient and left ventricular ejection fraction (LVEF) [1]: (i) AS with normal LVEF and high transvalvular gradient (NEF HGAS), (ii) AS with reduced LVEF and high transvalvular gradient (LEF HGAS), (iii) AS with reduced LVEF and low transvalvular gradient (LEF LGAS, also referred to as classical low-flow, low-gradient AS), and (iv) AS with normal LVEF and low transvalvular gradient (NEF LGAS, also referred to as paradoxical low-flow, low-gradient AS). These subtypes differ with respect to pathophysiological mechanisms, cardiac remodeling, and prognosis [7,8,9].

Myocardial remodeling occurs in various different myocardial diseases, including AS-induced hypertrophy and heart failure (HF), and involves alterations in cardiac energy metabolism [10,11]. These alterations are characterized by a shift of cardiac substrate utilization from free fatty acids to glucose and ketone bodies [12,13]. This results in an energetic deficit and thereby contributes to impaired LV function [14]. Several approaches to target substrate metabolism in HF are currently under preclinical evaluation [15,16], highlighting its relevance and therapeutic potential. By now, the understanding of metabolic changes in AS-induced remodeling in different hemodynamic subtypes is scarce. Metabolomics constitute a potent screening tool for defining changes in global and cardiac-specific metabolism in cardiovascular disease [17]. Therefore, we performed serum metabolomic profiling in patients presenting with different hemodynamic subtypes of AS as well as healthy blood donors (controls) in order to assess metabolic alterations in AS including the identification of subtype-specific changes. Our findings may help in the design of novel, targeted therapeutic strategies modifying cardiac metabolism to treat HF development in AS.

## 2. Materials and Methods

### 2.1. Patient Cohort

This study cohort consisted of 40 patients with severe AS who were scheduled for transfemoral transcatheter aortic valve implantation (TAVI), according to Puls et al. [18]. The indication for TAVI was based on heart team consensus according to current guidelines [1]. Transfemoral implantation was performed using standard techniques. In the vast majority of cases, the Sapien 3 valve (Edwards Lifesciences Inc., Irvine, CA, USA) was implanted. At baseline transthoracic echocardiography (TTE) and transoesophageal echocardiography, the 6-min walking test (6mwt), Minnesota Living with Heart Failure Quality of Life questionnaire (MLHFQ), New York Heart Association (NYHA) status, and N-terminal pro-brain natriuretic peptide (NT-proBNP) levels were recorded. Ten healthy blood donors (age: 60–70 years, 5 male and 5 female) served as controls. The local ethics committee approved the study and written informed consent was obtained from all patients and blood donors, respectively.

### 2.2. Echocardiography

All echocardiograms were performed using either a Philips ie33 or a Philips Epiq7 system, routinely recorded in a Picture Archiving and Communication System and retrospectively re-evaluated by a single observer using Q Station 3.8.5 (Philips healthcare). Echocardiographic measurements were obtained as recommended [19]. Following current guidelines, four subtypes of severe AS were defined:Normal/preserved ejection fraction, high-gradient AS (NEF HGAS): LVEF ≥ 50%, vmax ≥ 4 m/s or Pmean ≥ 40 mmHg, and aortic valve area (AVA) ≤ 1.0 cm^2^;Low/reduced ejection fraction, high-gradient AS (LEF HGAS): LVEF < 50%, vmax ≥ 4 m/s, or Pmean ≥ 40 mmHg, and AVA ≤ 1.0 cm^2^;Low/reduced ejection fraction, low-gradient AS (‘classic’low-flow, low-gradient AS) (LEF LGAS): LVEF < 50%, vmax < 4 m/s and Pmean < 40 mmHg, AVA ≤ 1.0 cm^2^, and stroke volume index (SVI) ≤ 35 mL/m^2^;Normal/preserved ejection fraction, low-gradient AS (’paradoxical’low-flow, low-gradient AS) (NEF LGAS): LVEF ≥ 50%, vmax < 4 m/s and Pmean < 40 mmHg, AVA ≤ 1.0 cm^2^ and indexed AVA ≤ 0.6 cm^2^/m^2^, and SVI ≤ 35 mL/m^2^. For this study, serum from 10 patients per subgroup plus 10 healthy controls were analyzed (Table 1).

### 2.3. Serum Sample Preparation

Global biochemical profiles were determined in human serum samples collected from healthy controls (blood donors) and patients prior to the respective TAVR procedure in our center. Cohorts were determined as shown below with grading by LVEF and transvalvular gradient. Venous blood samples were collected into procoagulant tubes under fasting conditions. Following centrifugation, the serum was transferred to another tube and was frozen at −80 °C. For metabolomic analysis, samples were thawed at room temperature for 20 min, vortexed and centrifuged at 650× *g* for 3 min before use. To remove protein, dissociate small molecules bound to protein or trapped in the precipitated protein matrix, and to recover chemically diverse metabolites, proteins were precipitated with methanol under vigorous shaking for 2 min (Glen Mills GenoGrinder 2000 [Glen Mills Inc., Clifton, NJ, USA]) followed by centrifugation. Samples were placed briefly on a TurboVap^®^ (Zymark, Hopkinton, MA, USA) to remove the organic solvent then reconstituted in compatible solvents before analysis. Finally, sample extracts were stored overnight under nitrogen before preparation for analysis.

### 2.4. Metabolomic Profiling

Metabolomic profiling was performed using ultrahigh-performance liquid chromatography/mass spectrometry (UHPLC/MS) in the positive (two methods) and negative (two methods) mode. Metabolites were then identified by automated comparison of ion features to a reference library of chemical standards followed by visual inspection for quality control as previously described [20]. Further details are given in the Appendix A. A list of all metabolite measurements in each individual sample is provided in Appendix A.

### 2.5. Statistical Analysis

For statistical analyses and data display, any missing values are assumed to be below the limits of detection; these values were imputed with the compound minimum (minimum value imputation). Statistical tests were performed in ArrayStudio (Omicsoft) or “R” to compare data between experimental groups; *p* < 0.05 was considered significant and 0.05 < *p* < 0.10 to be trending. An estimate of the false discovery rate (Q-value) is also calculated to take into account the multiple comparisons that normally occur in metabolomic-based studies, with q < 0.05 used as an indication of high confidence in a result.

The present dataset comprises a total of 1293 metabolites, 1009 compounds of known identity (named biochemicals), and 284 compounds of unknown structural identity (unnamed biochemicals). Following log transformation and imputation of missing values, if any, with the minimum observed value for each compound, Welch’s two-sample *t*-test was used to identify biochemicals that differed significantly between experimental groups. Serum metabolite levels were compared between all AS patients (referred to as All-AS group in this study) independent of their phenotype and controls as well as between the AS subgroups.

Principal component analysis (PCA) score plot was created using the statistical package programs R (http://cran.r-project.org/, accessed on 13 October 2022) [21]. Interactivenn (http://www.interactivenn.net/index2.html/, accessed on 13 October 2022) [22] was used to generate venn diagrams, Heat mapper (http://www.heatmapper.ca/, accessed on 13 October 2022) [23] to generate heatmaps. Metabolic pathway enrichment analysis and pathway topology analysis were conducted using the MetaboAnalyst 5.0 (https://www.metaboanalyst.ca/home.xhtml, accessed on 13 October 2022) [24] computational platform.

## 3. Results

Clinical and demographical characteristics of participants are displayed in Table 2.

Additional information on comorbidities and prescribed medication are listed in Appendix A.

### 3.1. Exploration with Principal Component Analysis and Hierarchical Clustering

In a first step, we assessed whether general metabolic regulation differed between the four AS subgroups and healthy controls (Figure 1A–C). In this regard, principal component analysis (PCA) resulted in an obvious separation of the control group, whereas the four different AS subtypes could not be clearly distinguished (Figure 1A). PCA revealed that one sample was clearly separated from all others and therefore was removed from further analyses. Similar metabolomic fingerprints of all four AS subtypes were further revealed by the heatmap in Figure 1B, presenting all significantly regulated metabolites in the AS subtypes vs. controls. The Venn diagram in Figure 1C shows the number of differentially regulated metabolites amongst the four AS subgroups compared to controls. Interestingly, the highest number of altered metabolites was found for the commonly regulated ones amongst all AS subgroups. Together, these findings indicate that AS appears to be associated with a universal metabolic disturbance and/or adaptation, independent of the respective hemodynamic subtype.

In the next step, we performed enrichment and pathway analyses to attain more mechanistic insights. We found ascorbate and aldarate metabolism, arginine biosynthesis, and arginine and proline metabolism to be the most significantly altered in the All-AS group compared to the controls (Figure 2A). Next, we analyzed the distinctly altered metabolites in each AS subtype separately. Pathway analysis highlighted a remarkable change of arginine biosynthesis, starch, and sucrose metabolism in the NEF HGAS group (Figure 2B). In the LEF HGAS group, the most affected pathways appeared to be the metabolism of valine, leucine and isoleucin (Figure 2C). Phenylalanine, tyrosine, and tryptophan biosynthesis was predominantly deranged in the NEF LGAS group (Figure 2D). No pathways were significantly enriched in the LEF LGAS group. This analysis indicates prominent changes in amino acids (AAs) metabolism in AS patients. This was associated with increased nitrogen load. When comparing the All-AS group to controls, numerous compounds involved in nitrogen balance, such as circulating urea, and polyamine levels were found to be significantly elevated (Appendix A), but no differences were found between the subgroups (Appendix A). Additionally, numerous amino acids were significantly reduced in the All-AS group vs. controls (Appendix A) with no differences between the subtypes (Appendix A).

### 3.2. Alterations in Acylcarnitine Levels

Oxidation of free fatty acids (FFAs) happens primarily in the matrix of the mitochondria. Delivery of FFAs begins with the conjugation of FFA with CoA and subsequently carnitine. Acylcarnitines are then transported across the mitochondrial membranes (Figure 3A). Following import, FFAs can then be oxidized to produce acetyl-CoA and enter the TCA (tricarboxylic acid) cycle to generate cellular energy. In this study, nearly all measured acylcarnitines were elevated in All-AS compared to controls, while no significant differences were detected between different subgroups (Appendix A), as exemplarily shown for palmitoylcarnitine and myristoylcarnitine (Figure 3B–D).

### 3.3. Alterations in Ketone Body Metabolism

Circulating ketone bodies serve as an alternative fuel source. Under energetic demand, fasting, and carbohydrate restriction, they are taken up by peripheral organs, and rapidly converted to acetyl-CoA to enter the TCA cycle (Figure 4A). 3-Hydroxybutyrate (BHBA) is the predominant circulating ketone body. Here, we observed >10-fold increase (*p* < 0.05) in circulating BHBA in the All-AS group compared to the control group (Figure 4B,C). In addition, several other side products of ketone metabolism e.g., 3-hydroxy-3-methyglutrate and the less common ketone body acetoacetate, were also elevated in the All-AS group (Figure 4,B,D). There was no significant difference between the AS subgroups (Figure 4C,D, Appendix A).

### 3.4. Alterations in Sugar Metabolism

In the All-AS group, we noted elevated serum levels of numerous metabolites involved in sugar metabolism including the pentose phosphate pathway (PPP), glycogen and amino sugar metabolism when compared to controls, indicating a metabolic shift towards glucose utilization. Of note, no alterations in pyruvate serum levels in the All-AS group could be observed. In line with our previous findings, these metabolites did not reveal any differences between the AS subgroups (Figure 5A–D, Appendix A).

### 3.5. Differences in Metabolite Levels between AS Subgroups

In this study, the LEF HGAS group had elevated circulating levels of all detected hemoglobin metabolites (e.g., bilirubin) compared to healthy controls. Specific group comparisons revealed particularly enhanced levels of hemoglobin metabolites in the LEF HGAS group when compared to the other AS subgroups (Figure 6A, Appendix A).

In addition to the general changes in lipid metabolism in AS patients, we identified more abundant diacylglycerols (DAGs) in the NEF HGAS group compared to the control group as well as to the other AS groups (Figure 6B, Appendix A).

Dihydrosphingomyelins (DHSMs) are a subclass of sphingomyelins, in which the sphingosine backbone is hydrogenated. Here, we observed low levels of DHSMs in AS patients with reduced LVEF, i.e., the LEF HGAS and LEF LGAS (Figure 6C, Appendix A).

## 4. Discussion

To the best of our knowledge the present study is the first to evaluate metabolic remodeling in different hemodynamic subtypes of AS using a serum metabolomic approach.

The key findings of this investigation are: (i) Serum metabolomic profiles of all AS groups combined show significantly altered metabolite levels, e.g., acylcarnitines and ketone bodies, involving different metabolic pathways vs. healthy controls; (ii) no relevant differences in acylcarnitine, ketone or sugar metabolism could be observed between the AS subtypes; and (iii) several different metabolites were differentially regulated between AS subgroups, including bilirubin, diacylglycerols and dihydrosphingomyelins.

Changes in cardiac substrate utilization are part of the myocardial remodeling process that occurs in various cardiac diseases such as hypertensive heart disease, dilated cardiomyopathy and coronary artery disease [10,11,12,13]. Cardiac hypertrophy arises as an adaptive response to AS-induced afterload; however, sustained hemodynamic overload leads to maladaptive remodeling and consequently HF [25,26]. Experimental studies of pressure overload-induced hypertrophy also revealed altered cardiac metabolism [27].

Using a novel in vivo labeling methodology, we recently demonstrated increased cardiac glycolysis, glucose-dependent TCA cycle activity, anaplerosis and de novo glutamine synthesis in a murine model of pressure overload. Interestingly, we did not find these changes during volume overload, suggesting that metabolic remodeling largely depends on the particular stressor/stimulus [28]. During the last years, serum metabolomic profiling has emerged as useful tool to better understand metabolic processes in cardiac diseases [29,30,31]. In our study, we identified a variety of different metabolites that were commonly regulated in all AS subgroups indicating generic mechanisms irrespective of the hemodynamic subtype, as well as subtype-specific changes.

### 4.1. Acylcarnitines

Under physiological conditions, fatty acids provide 70% of the fuel requirements to the heart. Acylcarnitines are used to transport fatty acyls into the mitochondria and therefore play a role in providing substrates derived from fatty acid uptake to cardiomyocytes [32]. Since the serum lacks the cellular machinery for FFA metabolism, serum levels of acylcarnitines often reflect levels within heart tissue [33] and changes in serum levels may reflect impaired cardiovascular function [34]. Decreased levels of long chain acylcarnitines after TAVR or treatment of decompensated HF were linked to improved LV function and might act as a marker for reverse remodeling [30,35,36]. Measuring acylcarnitine levels before and after TAVR might be helpful to identify patients at risk for poor outcome.

Here, we observed an increase in nearly all groups of acylcarnitines in all AS subgroups. This observation may be due to two different reasons: Firstly, elevated serum acylcarnitines could result from increased energetic demand being met by fatty acid oxidation [37,38]. This explanation is based on different studies which reported increases in FFA uptake as well as in acylcarnitines in patients with AS or HF with preserved systolic function [29,39,40]. Secondly, ineffective fatty acid oxidation leads to an excess of upstream metabolites including acylcarnitines [41]. This is consistent with the general concept of defective fatty acid import into cardiomyocytes in HF with reduced LVEF (HFrEF), especially in end-stage HF [42]. Interestingly, we did not observe different acylcarnitine levels between the different AS subtypes. One may speculate that in the more compensated NEF HGAS group, the increase may reflect a higher demand for active fatty acid oxidation in the heart, whereas in more decompensated stages with reduced LVEF (i.e., LEF HGAS and LEF LGAS), the increase in acylcarnitines may be a result from defective fatty acid oxidation and accumulation of upstream metabolites. This needs further clarification through future mechanistic studies.

### 4.2. Ketone Metabolism

In pathological conditions, when fatty acid oxidation is diminished, the heart is able to utilize other substrates to meet its energy demand. Various studies previously reported an increased use of ketone bodies for energy production in HF [43,44]. Interestingly, this correlates with increased blood ketone body levels in HF patients [45]. Voros and colleagues observed not only increased plasma levels of ketone bodies in patients with AS, but also an augmented myocardial uptake of these circulating ketone bodies [39]. The increased use of ketone bodies appears to constitute an early mechanism of the heart to meet elevated energy demand. In line with these findings, an experimental study in mice with pressure induced HF already showed increased ketone body oxidation before the development of overt HF [44]. However, whether this alteration in ketone body utilization represents an adaptive or maladaptive mechanism remains elusive. In our study, we similarly observed augmented circulating ketone bodies in patients with AS irrespective of the LVEF and transvalvular gradient. Together with the observed changes in acylcarnitines, this points towards increased energetic demand and a related change in substrate utilization for energy production in AS patients. Similarly, as previously discussed for acylcarnitines, the increase in ketone bodies was not different between the different AS subtypes, indicating a generic mechanism.

### 4.3. Sugar Metabolism

In pathological cardiac remodeling, there is a prominent metabolic shift towards glucose utilization to compensate for the reduction in fatty acid oxidation [46]. Our results did not show increased glycolysis, including anaerobic glycolysis, in our patients as indicated by unchanged serum levels of pyruvate and lactate. In this context, some studies also reported decreased rates of cardiac glucose uptake and glycolysis as a consequence of myocardial insulin resistance [47,48]. As diabetes was highly prevalent in our AS patient cohort, this might explain why we did not observe changes in glycolytic metabolites such as pyruvate. Moreover, as pyruvate is highly instable in serum this could serve as an alternative explanation for the unchanged levels of this metabolite and not reflect physiological mechanisms.

Importantly, increased glucose levels in our AS patients rule out that the observed changes in acylcarnitines and ketone bodies are due to fasted state of the respective patients. The pentose phosphate pathway (PPP) is an accessory pathway of glucose metabolism, which plays an important role in maintenance of cytosolic redox homeostasis [49]. Studies showed that increase glucose flux through the pentose phosphate pathway (PPP) reduces reactive oxygen species in the heart and subsequently attenuates hypertrophy [50]. Here, we observed augmented serum metabolites of the PPP indicating altered PPP metabolism in AS patients. However, other studies suggest that excessive NADPH derived from PPP might contribute to the development of HF [51]. Therefore, it remains unclear whether the increased PPP metabolites have beneficial or detrimental effects on AS-induced remodeling.

### 4.4. Amino Acid Metabolism and Nitrogen Balance

On its search for alternative fuels in disease, the heart is also able to metabolize amino acids. Especially glutamate, glutamine, aspartate, asparagine and the branched chain amino acids (BCAAs) have been shown to be preferentially used as metabolic substrates in the Krebs cycle [52,53,54]. Accordingly, reduced peripheral levels of amino acids, as observed in our All-AS group, may indeed reflect increased cardiac amino acid utilization. If they are being catabolized for energy production, this would in turn increase the need to eliminate excess nitrogen, and potentially lead to the observed increases in circulating urea and polyamine levels. However, renal dysfunction, as evident in our AS patients, may also be a strong contributor to these increases which needs to be considered.

### 4.5. Observed Biochemical Changes (AS Subtyping)

#### 4.5.1. Hemoglobin Metabolism

Circulating bilirubin was found to be more sensitive to hemodynamic disturbances than liver transaminases and its increase appeared to be coincident with cardiac decompensation in chronic HF [55,56]. Additionally, serum bilirubin was shown to have a negative inotropic effect despite its antioxidant and anti-atherogenic properties [57,58]. Hosoda et al. also reported poor prognosis in HF patients with elevated bilirubin levels undergoing cardiac resynchronization therapy compared to HF with low bilirubin levels [59]. In the present study, increased serum bilirubin levels in LEF HGAS compared to the NEF HGAS patients might indicate the transition of AS-induced HF from a well compensated to a decompensated stage. LEF HGAS patients showed a significantly lower mortality rate and favorable outcome after TAVI compared to LEF LGAS patients [60], despite harboring higher serum bilirubin levels. This might be, at least to some extent, related to the protective effects of bilirubin on the cardiovascular system. The potential of serum bilirubin as a predictive biomarker in AS needs to be assessed in future studies.

#### 4.5.2. Diacylglycerols (DAGs)

DAGs are produced during triglyceride synthesis and decomposition. Accumulation of lipid intermediates has also been implicated in the development of cardiac dysfunction and HF (i.e., lipotoxicity). Mechanistically, DAGs activate several isoforms of protein kinase C that have been linked to the development of pathological cardiac remodeling, including hypertrophy, fibrosis, and inflammation [61]. A rodent model of spontaneous hypertension showed increase in DAG content in early stage of the disease concomitant with the induction of cardiac hypertrophy suggesting that DAG accumulation appears to be related to initial hypertrophy. This was followed by a decrease in the DAG content and in the de novo fatty acid synthesis at late stage of the disease [62]. Consistently, our results showed elevation of DAG levels in compensated NEF HG AS patients, but a significant decrease was observed in patients with severe cardiac dysfunction, i.e., in LEF HGAS. DAGs may therefore also represent an interesting biomarker during the HF transition process.

#### 4.5.3. Dihydrosphingomyelins (DHSMs)

Sphingomyelins (SMs), a subclass of sphingolipids, are enriched in the outer-leaflet of the cell membrane and are particularly rich in various specific structures including the lens of the eye and the myelin sheath of neurons [63]. Emerging evidence suggests that plasma SMs are correlated with LV systolic dysfunction [64]. Another study showed that the effect of plasma SMs on the cardiac function depends on their species. While higher plasma levels of palmitic acid enriched SMs were associated with higher risk of incident HF, SMs with a very long-chain saturated fatty acids were associated with lower risk of HF. Associations appeared similar with the outcomes of HF with preserved ejection fraction (HFpEF) and HFrEF [65]. Interestingly, decreased myocardial DHSMs were detected in a mouse model of myocardial infarction [66] suggesting a protective role of DHSMs in the heart. Consistently, increased DHSMs predict slower disease progression in patients with Alzheimer’s [67]. Whether depletion of plasma DHSMs in AS patients with reduced EF, as observed in our study, is associated with the deterioration of cardiac function in AS patients is still unexplored and needs further investigations.

### 4.6. Study limitations

(i) It must be noted that analyses in serum, as carried out in this study, may not necessarily reflect processes in the heart but may be affected by other organs. Therefore, this study design only allows to generate hypotheses, but not causal relationships between metabolite expression and clinical features. Hence, it is speculative as to which metabolites are most promising as diagnostic and/or prognostic biomarkers in AS, particularly in different hemodynamic AS subtypes. (ii) Sample size within the AS subtypes was relatively small with 10 patients per group. Thus, the observed changes need to be confirmed in larger longitudinal study patient cohorts with serial sampling, ideally before and after TAVI to evaluate prognostic implications. (iii) The greatest differences in serum metabolomic profiles were detected between healthy controls and AS patients. Notably, the group of AS patients was older than the controls, and this could clearly have an effect on the results. (iv) Comorbidities and coexisting age-related health issues are very common in AS patients and they strongly influence the systemic and myocardial metabolism [68,69]. Therefore, they are likely to affect our results. In such instances, molecular explorations in experimental animal models mimicking the human disease state would be important to confirm metabolic alterations observed in AS patients and to better understand their role on a mechanistic level. Additionally, the concentration of serum metabolites can be affected by impaired renal elimination. Indeed, we found elevated levels of creatinine and urea in AS patients compared to controls. (v) All patients were fasting, and blood samples were collected under similar conditions. However, data on specific dietary patterns and lifestyle parameters that might influence patients’ metabolic profile were lacking. (vi) The effect of specific medications (e.g., anti-diabetic or anti-dyslipidemic) prescribed to our patients on serum metabolites could not be analyzed in detail in this study.

With respect to these confounding factors, mentioned under (iv), (v) and (vi), larger studies are direly need to precisely assess their contribution towards metabolic changes in AS including different hemodynamic subtypes.

## 5. Conclusions

Despite several limitations, our study still provides relevant insights into AS-induced metabolic alterations, both subtype-specific as well as across all subtypes. In this regard, numerous serum metabolites (e.g., acylcarnitines, ketone bodies, amino acids) and associated metabolic pathways (e.g., arginine biosynthesis and ascorbate and aldarate metabolism) were identified to be differentially regulated in AS. These did not significantly differ between the four hemodynamic subtypes of AS, suggesting generic mechanisms. Additionally, metabolites involved in hemoglobin and lipid metabolism (e.g., bilirubin, dihydrosphingomyelins, and diacylglycerols) were found to be regulated AS subtype-specific, suggesting distinct metabolic regulation. From a diagnostic and prognostic point of view, the clinical implications of these findings remain an interesting and important avenue for future research.

## Figures and Tables

**Figure 1 biomolecules-13-00095-f001:**
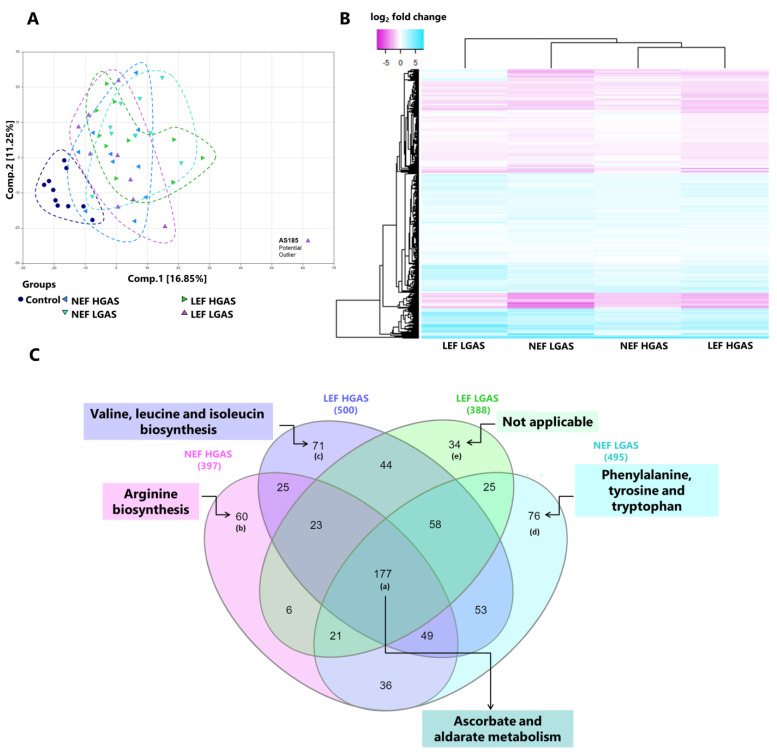
Metabolomic fingerprints in AS subgroups and healthy controls. (**A**) Principal component analysis (PCA). (**B**) Heat map representing hierarchical clustering of differentially abundant metabolites. Metabolites were ordered by euclidian distance and average linkage according to the ratios of abundance in each AS subgroup when compared to controls. Each row represents an individual metabolite, and each column represents patient group. Pseudocolors indicate differential abundance (blue, pink, white representing metabolite levels above, below or equal to the mean on a log_2_ scale, respectively). (**C**) Venn diagram showing differentially regulated metabolites and pathways shared or unique among the different groups. The numerical values on the Venn diagram depicted the number of regulated metabolites.

**Figure 2 biomolecules-13-00095-f002:**
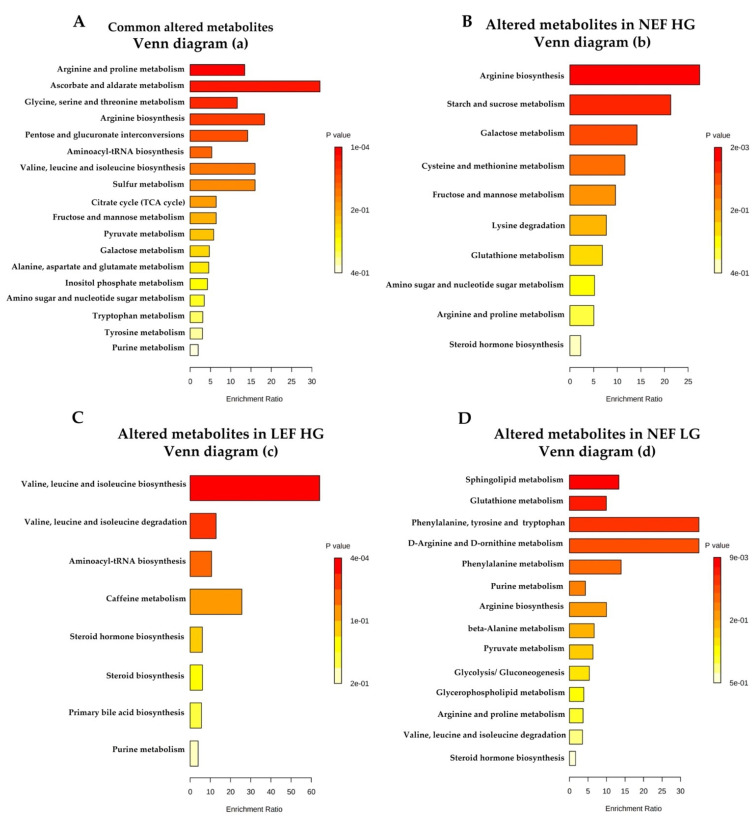
Pathway analysis. (**A**) The most altered functional metabolic pathways among all AS subgroups. (**B**–**D**) Unique altered functional metabolic pathways in each AS subgroup. The enrichment analysis was implemented using the hypergeometric test to evaluate whether a particular metabolite set is represented more than expected by chance within the given compound list. One-tailed *p* values are provided after adjusting for multiple testing.

**Figure 3 biomolecules-13-00095-f003:**
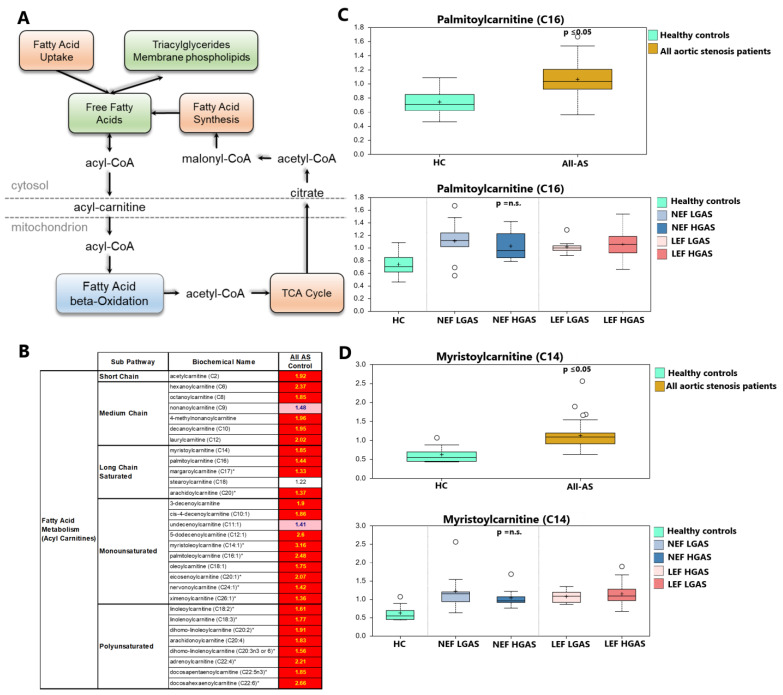
Altered metabolism of fatty acids. (**A**) Illustration of fatty acid uptake to the mitochondria by transfer of acyl-residues to acylcarnitine. (**B**) List of altered aclycarnitines between AS patients and healthy controls (significantly regulated molecules marked in red). Values marked in light red indicate a trend towards statistical significance (0.05 < *p* < 0.10). Non-colored values are not significantly different for that comparison. Numbers indicate x-fold increase in concentration. * *p* ≤ 0.05. (**C**,**D**) Palmitoylcarnitine (C16) and myristoylcarnitine (C14) as examples between All-AS vs. controls as well as the different AS subgroups. On each box, the black cross indicates the mean value, the borders and segmentation indicate limits of upper/lower quartile and median values. Outliers are plotted as circles. n.s.: not significant between AS subgroups.

**Figure 4 biomolecules-13-00095-f004:**
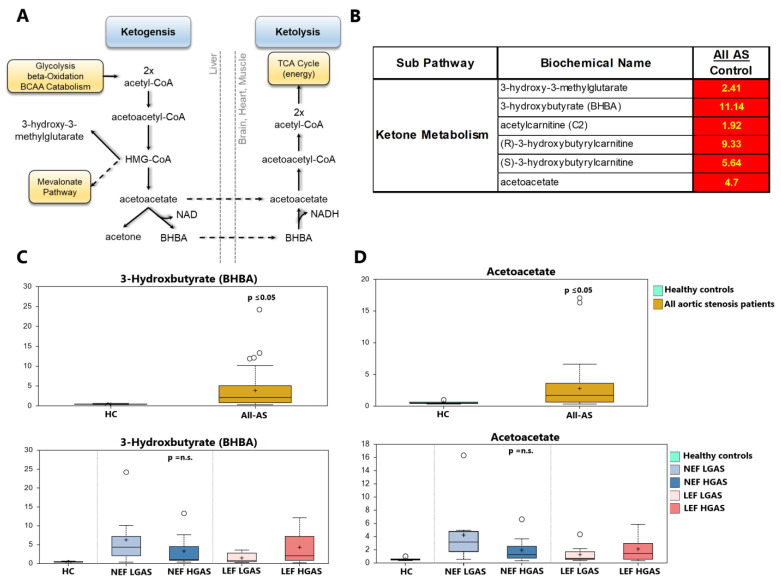
Ketone bodies in AS and healthy controls. (**A**) Illustration of ketone body utilization for energy production via TCA cycle. (**B**) List of significantly altered ketone bodies between All-AS group and controls (significantly regulated molecules marked in red, numbers indicate x-fold increase in concentration). (**C**,**D**) 3-Hydroxbutyrate (BHBA) and acetoacetate as examples between All-AS vs. controls as well as the different AS subgroups. On each box, the black cross indicates the mean value, the borders and segmentation indicate limits of upper/lower quartile and median values. Outliers are plotted as circles. n.s.: not significant between AS subgroups.

**Figure 5 biomolecules-13-00095-f005:**
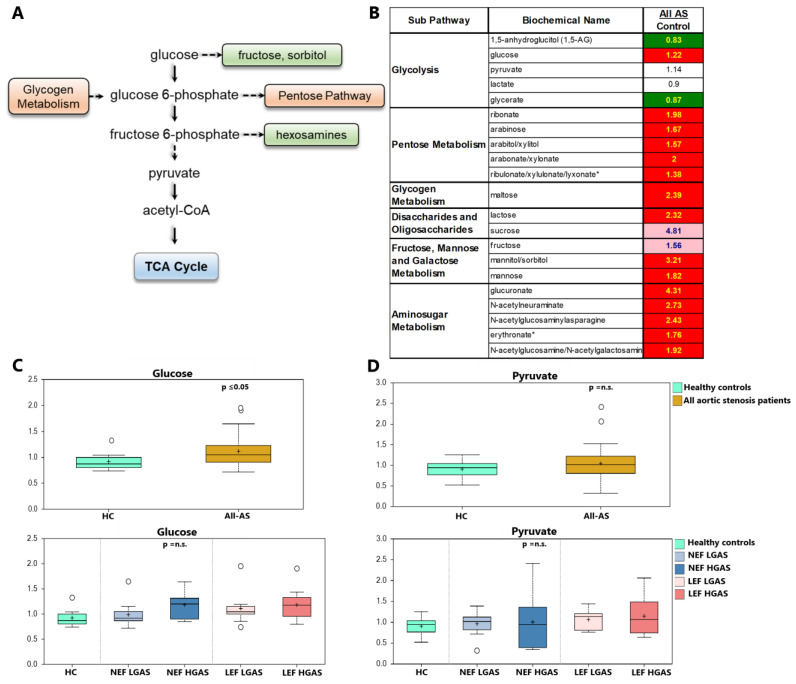
Sugar metabolites in AS and healthy controls. (**A**) Illustration of glucose utilization pathways for energy production via TCA cycle including glycolysis, pentose phosphate pathway (PPP), and hexosamine biosynthetic pathway. (**B**) List of altered metabolites. Significantly up- and downregulated molecules are marked in red and green, respectively. Values marked in light red indicate a trend towards statistical significance (0.05 < *p* < 0.10). Non-colored values are not significantly different for that comparison. Numbers indicate x-fold increase/decrease in concentration. * *p* ≤ 0.05. (**C**) Glucose as an example between All-AS vs. controls as well as the different AS subgroups. (**D**) Pyruvate as an example between All-AS vs. controls as well as the different AS subgroups. On each box, the black cross indicates the mean value, the borders and segmentation indicate limits of upper/lower quartile and median values. Outliers are plotted as circles. n.s.: not significant between AS subgroups.

**Figure 6 biomolecules-13-00095-f006:**
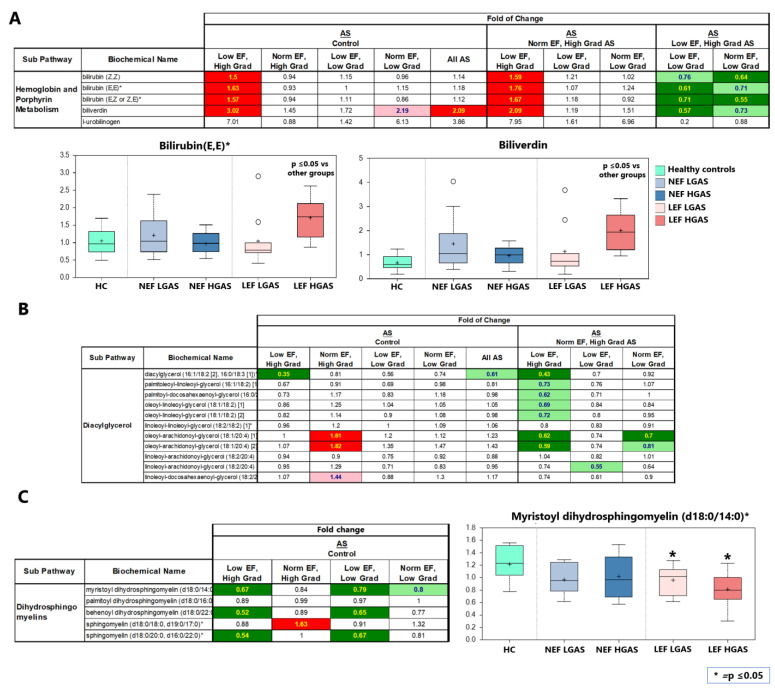
Metabolites with different serum levels between AS subgroups. (**A**) List of altered hemoglobin related metabolites. Bilirubin and biliverdin are illustrated as examples. (**B**) List of altered diacylglycerols (DAGs) in AS patients. (**C**) List of altered dihydrosphingomyelins (DHSMs) in AS subgroups. Myristoyl dihydrosphingomyelin is illustrated as an example. Significantly, up- and downregulated molecules are marked in red and green, respectively. Values marked in light red and light green indicate a trend towards statistical significance (0.05 < *p* < 0.10). Non-colored values are not significantly different for that comparison. Numbers indicate x-fold increase/decrease in concentration. On each box, the black cross indicates the mean value, the borders and segmentation indicate limits of upper/lower quartile and median values. Outliers are plotted as circles. Asterix (*) indicates *p* ≤ 0.05 between the annotated group and the control group.

**Table 1 biomolecules-13-00095-t001:** Graduation of different subtypes of AS as determined by the ESC guidelines; LVEF was determined as ≤50%, HG AS was determined as gradient ≥40 mmHg.

Group ID	n	Description
Ctrl	10	Healthy controls
NEF HGAS	10	Normal EF, high gradient AS
NEF LGAS	10	Normal EF, low gradient AS
LEF HGAS	10	Low EF, high gradient AS
LEF LGAS	10	Low EF, low gradient AS

**Table 2 biomolecules-13-00095-t002:** Study participants’ characteristics. 40 patients prior to TAVR (10 patients per AS subgroup) and 10 healthy controls were included in this study. BMI: body mass index, LVEDD: left ventricular end-diastolic diameter.

	Control	NEF HGAS	LEF HGAS	LEF LGAS	NEF LGAS
Demographics					
Male gender, n (%)	5 (50%)	5 (50%)	6 (60%)	9 (90%)	6 (60%)
Age (years)	60–70	79.4 ± 4.1	77.1 ± 9.0	79.5 ± 6.4	80.4 ± 5.9
BMI		29.66 ± 5.2	28.31 ± 8.2	25.82± 3.9	30.38 ± 7.3
Laboratory measures					
Hemoglobin (g/dL)		13.2 ± 1.9	13.1 ± 2.2	12.7 ± 1.6	11.8 ± 1.9
Creatinine (mg/dL)		1.27 ± 0.41	0.97 ± 0.14	1.13 ± 0.48	1.23 ± 0.40
NT-proBNP (ng/L)		2717 ± 4412	10964 ± 12751	7069 ± 7388	2210 ± 1317
Echocardiography					
LVEF (%)		63.4 ± 8.8	30.1 ± 9.1	33.3 ± 7.1	61.9 ±7.6
LVEDD (mm)		40. 7 ± 3.3	54.6 ± 9.6	51.8 ± 7.3	43.1 ± 5.6
Aortic valve orifice area (cm^2^)		0.70 ± 0.16	0.59 ± 0.11	0.71 ± 0.18	0.77 ± 0.14
Peak gradient (mmHg)		52.9 ± 10.4	48.4 ± 11.3	26.7 ± 5.0	26.3 ± 5.1

## Data Availability

Data are stored in a database provided by the company Metabolon^®^. All raw data are available in Appendix A.

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
