# Peer review of "Metabolomic Profiling in Patients with Different Hemodynamic Subtypes of Severe Aortic Valve Stenosis"

_biomolecules, 2023, doi:10.3390/biom13010095_

Round 1
Reviewer 1 Report
Many thanks for the opportunity to review this work. I commend the authors for their efforts in performing detailed metabolic profiling of AS patients. Unfortunately, the authors have failed to address/ discuss the practical implications (clinical diagnostic/prognostic values, target identification for drug developments etc.) sufficiently.
Literature is replete with examples of metabolic profiling of AS and/or HF patients (e.g., Metabolic Dysfunction in Heart Failure: Diagnostic, Prognostic, and Pathophysiologic Insights From Metabolomic Profiling - PubMed (nih.gov)). As such, for this piece to have made a meaningful contribution to the existing knowledge base, the authors should have discussed their findings in the context of 'potential application to future clinical practice'. As it stands, much of the paper is merely describing the observations of the omics analysis and superfluous comparisons to previous studies that report similar results.
Reviewer 2 Report
To:
Editorial Board
Biomolecules
Title: “Metabolomic profiling in patients with different hemodynamic subtypes of severe aortic valve stenosis”
Dear Editor,
I read this paper and I think that:
- The results section of the abstract is poorly written. Authors should include more numerical data in order to improve the readability of the text.
- The methods section of the main manuscript is ambiguous. Authors stated that they included 200 patients with TAVI but the final number is 40. I could understand the description of the background but the paragraph should be re-written for the sake of clarity.
- Authors reported did not report any data on pharmacological treatment. Pharmacological treatments might impact on results. This is a great limitation of the paper which should be addressed.
- The role of dietary habits might also impact on metabolomics. Nutraceuticals are able to influence the metabolism of glucose and lipids (see also Scicchitano P et al. Journal of Functional Foods 2014;6:11-32). This point should be discussed and well addressed in the text.
- There is no mention about the comorbidities of patients. These might represent a further confounding factor in the final analysis. Please address this comment into a dedicated limitation section
Reviewer 3 Report
The topic of the study of this metabolomic profiling approach is of high interest to readers of Biomolecules. This manuscript includes very interesting data, which provide novel insights into
pathophysiological mechanisms of different subtypes of severe aortic stenosis. In my opinion, in general, the experiments are well designed and the text is well written, the data is very well presented, so, there are only a few minor considerations that need to be corrected.
Minor aspects:
· It is not clear to me why out of 200 participants only used 40 + 10 healthy controls. It would be advantageous to have a larger number of participants in each group.
· What is the percentage of missing data obtained? Must refer.
· Table 1. Is it possible to indicate the BMI and laboratory measures for the controls, or other pertinent parameters?
· From what I understand authors did Welch's two-sample t-test for the 1293 metabolites for each comparison. Why didn't authors do a PLS-DA pairwise analysis for example and only evaluate the potentially discriminating metabolites? The number of multiple comparisons would be considerably reduced as well as the work
